# Challenges and Lessons from an Acute Telehealth Homeopathy Service During the Pandemic: A Case Series Exploring How Changing Demographics, Efficiency and Outcomes, Point to New Options for Epidemic Readiness

**DOI:** 10.3390/healthcare13010003

**Published:** 2024-12-24

**Authors:** Alastair C. Gray, Parker Pracjek, Christine D. Luketic, Denise Straiges

**Affiliations:** HOHM Foundation, Philadelphia, PA 19138, USA

**Keywords:** homeopathy, complementary therapies, complementary medicine, complementary and alternative medicine, organizational change, epidemic preparedness, epidemic readiness

## Abstract

**Background:** In 2020, HOHM Foundation launched Homeopathy Help Now (HHN), a network of professional homeopathy telehealth practitioners, administrative volunteers, and independent researchers to work collaboratively in order to respond to the urgent need of care for the ever-growing number of COVID-19 cases in the United States. **Methods:** in this pragmatic case series study, cases of positively testing or probable COVID-19 (n = 3495) are analyzed using conventional quantitative analysis. The sample includes clinical data collected from clients who attended the clinic between 23 March 2020 and 31 December 2023. **Results:** The youngest client at the clinic was less than one year old, and the eldest was 92. Many of the participants at this clinical facility were adults (58.1%), with fewer (41.9%) aged 0–17. Many were female (61.7%), while fewer were male (32.4). Most clients found their symptoms improved at final contact (83.6%), and the majority of individual remedy responses improved symptoms (73.7%) over the course of their care. **Discussion:** Health Services and Public Health research projects are warranted to investigate the ways in which such a necessary stop-gap clinical service as HHN could become implemented in early- and later-phase response to pandemics. HHN’s collaborative, horizontally integrated team structure was essential for the creation of the novel approach needed to address the serious symptoms of COVID-19. Moreover, HHN’s organizational model draws on a mutual aid structure, whereby dynamic, flexible systems are created that empower a community to meet emerging needs, especially when more formal structures are strained, failing or simply unavailable. Further research is urgently needed into the implementation and benefits of innovative, flexible healthcare structures, such as the one used in this study, that can meet the unpredictable and fluctuating public health needs in our changing world.

## 1. Introduction

Homeopathy is one of the whole healing systems that make up complementary medicines [1] (CM)—here defined as healthcare not traditionally associated with the conventional medical profession or medical curriculum [2] and which includes a diverse field of treatments (e.g., reflexology, aromatherapy), mind–body practices (e.g., meditation, yoga), natural products (e.g., supplements, herbal medicines) and systems of medicines (e.g., Ayurveda and homeopathy) [3,4].

Individualized treatment using homeopathy is intended to stimulate a self-healing response in order to reduce or remove symptoms and help an individual move toward a more resilient, more dynamic state of health [5]. However, in this system of healing, homeopaths when confronted with individual illnesses associated with epidemic/pandemic disease, rather than deal with those cases as they would in everyday individualized practice, combine symptoms of a large group of people to define the ‘*genus epidemicus*’ of the disease—a distinct, small cluster of homeopathic remedies which are able to match the larger pattern of the disease [6] and which inform clinical decisions. The system of homeopathy has clear principle-based guidelines for the control of infectious epidemics, and these were established long before the advent of modern sanitation, vaccination, and antibiotics, with the first documented use in the management of scarlet fever in 1799 [7]. Homeopathic medicines have been used in the resolution of many epidemics [8], including diphtheria, typhoid, cholera [9], smallpox, Spanish flu, yellow fever, scarlet fever, [10], polio and meningitis, the success of which contributed to the ever-growing popularity of homeopathy in the USA and Europe in the nineteenth and twentieth centuries [11].

In 2020, HOHM Foundation [12] launched Homeopathy Help Now (HHN) [13], a network of professional homeopathy administrative volunteers, independent researchers and telehealth practitioners, to work collaboratively to respond to the urgent healthcare needs for the daily growing number of cases of COVID-19 in the United States. At the time, public health services were being inundated, and, at the frontline, many professional homeopaths were approached by clients who had been turned away from or were otherwise unable to access conventional healthcare services. The goal at HHN was to make a meaningful impact on the health of as many people as possible through the application of best practices according to the principles of homeopathy while maximizing such modern digital capabilities as remote telehealth access to care and communication with clients and stream-lined record-keeping and data collection, alongside working very closely as a team via virtual meeting platforms. In 2021, HHN expanded its telehealth services to include non-COVID 19-presenting acute illnesses seen through the teaching clinic of the Academy of Homeopathy Education [14].

Alongside the immediate clinical initiative, HOHM Foundation implemented a data collection protocol to capture as many clinical data as possible. In the early days of the pandemic, a variety of clinical advice and early research about homeopathy was found in the published literature involving a wide array of methods and results. The justification for this research is to identify and analyze data resulting from the use of the traditional homeopathy *genus epidemicus* methodology during the pandemic in one clinical setting. The aim of the study is to use clinical findings on the use of homeopathy and the *genus epidemicus* to further understand best practices. The objectives of the study are to highlight measures of demographics, effectiveness and efficiency.

## 2. Materials and Methods

### 2.1. Study Design

Cases of positively testing or probable COVID-19 (n = 3495) are analyzed using conventional quantitative analysis. At the initial onset of the pandemic, testing was not an option. Probable COVID-19 was defined by the symptoms presented in each case and documented. Further aspects of the study design are described below.

### 2.2. Setting

The setting of this study is Homeopathy Help Now [13], a coalition of professional practitioners and students working to further the mission of HOHM Foundation [12], which endeavors to provide accessible healthcare using homeopathy. Clinical provision is via a telehealth service [15], and no in-person consultations are offered.

### 2.3. Participants

The sample in this study includes clinical data collected from clients who attended the clinic between 23 March 2020 and 31 December 2023 [16]. Self-selecting individuals seeking assistance for their probable COVID-19 or COVID-19 positive symptoms navigated to the portal on the HHN website [17].

### 2.4. Variables

Data were collected through the online intake form that includes name and further demographic details (including date of birth, gender, client location, pre-existing conditions, current work with a homeopath, access to remedies, familiarity with homeopathy, preferred method of communication and referral pathway), as well as a synopsis of symptoms, the date illness began, the nature of the health concern, and any conventional medications that may have been taken for the acute condition. Additionally, all clients were required to sign a legal consent form in which they affirmed their agreement to be consulted by a professional practitioner of homeopathy. Following a consultation (which was conducted over the phone or through video conferencing software such as Zoom (version 6.2.11) as appropriate to the client’s condition and accessibility to technology), ongoing clinical data were compiled into case notes during the consultations. This information includes details of the presenting complaint, COVID-19 status at intake (tested positive, tested negative, suspected due to case symptoms, suspected due to known exposure to COVID-positive person), any other types of practitioners or healing modalities used, co-morbidities, clinical outcome at final contact, individual remedy outcome, and prescription details of the actual remedies used. Data continued to be collected at each follow-up consultation until the close of the case.

### 2.5. Data Sources/Measurement and Analysis

When cases were closed, practitioners then submitted cases to the research data team for review, analysis, and processing. At that point, any incomplete data were identified, and clarification from individual practitioners was sought. Cases were coded for de-identification, a process that included the allocation of code assigned to each practitioner, as well as the location (country and state) of each practitioner. De-identified data were uploaded to an HHN portal. The final analytical sample emerged after a process of data cleaning that included duplicate removal, a process for cross-checking for the accuracy of data entry and adding missing data from clinical case notes.

### 2.6. Study Size

A sample of 3495 cases who attended the clinic in 2023 were analyzed.

### 2.7. Quantitative Variables

Many factors impacted the data inclusion and exclusion criteria in this study. These factors continued to evolve over the course of the pandemic as more definitive information about the pandemic became disseminated. For example, during the early days of the pandemic, widespread challenges and limitations to COVID-19 testing hindered diagnostic confirmation—especially in those situations where no known point of exposure existed. Further challenges emerged as testing became more readily available, as did the questionable reliability of those tests [14,15] which complicated clear diagnoses, as well as unfolding questions about immunity, re-infection, and the continually emerging breadth of symptoms and body systems impacted by the virus. Individuals whose symptoms were clearly outside the recognized COVID-19 cluster of symptoms and were determined to be of a chronic nature were referred to the HHN clinic for chronic care or appropriate healthcare providers.

### 2.8. Ethical Clearance

Ethical approval was granted by the Research Ethics Board of the Canadian College of Naturopathic Medicine, in 2024 (CCNMREB053.Gray.Pracjek, approved on 10 April 2024).

### 2.9. Statistical Methods

Descriptive statistical analysis (undertaken in April 2024) was employed including frequencies and percentages for categorical variables and means as well as standard deviations for continuous variables. Associations between categorical and continuous variables were examined using *t*-tests. Mixed-methods analyses were also conducted using Excel and the statistical software JMP Statistics (JMPPro16).

## 3. Results

The results are presented in three clusters: demographics, the efficiency of the service, and measurements of effectiveness.

### 3.1. Demographics

Findings (presented in Table 1) show that the youngest client at the clinic was less than one year old and the eldest was 92 years of age. Most participants were adults (58.1%), with fewer (41.9%) aged 0–17. Most participants were female (61.7%), while fewer were male (32.4%) Several clients identified as neither male nor female, or as transgender. There were insufficient numbers to share these numbers without jeopardizing client anonymity. Thirty-six point seven percent of participants reported that they currently had COVID-19 adjacent symptoms, while 32.5% reported that the question was not applicable to the case. Sixteen percent reported a positive test, and 6.7% reported a COVID-19 negative test result. Additionally, 3.5% of participants reported previously testing positive, while 2.1% reported testing positive during time of care, after the initial request was made for treatment. Most clients had never seen a homeopath before or used homeopathy (81.2%).

### 3.2. Efficiency

For the duration of the pandemic, despite the clinical quality of care, privacy, technical barriers, and security challenges, data continued to be collected alongside clinical care. A review of data (Table 2) related to the efficiency levels of the HHN administrative team, including systems for processing client requests as well as practitioner assignment, demonstrates that the time from form submission to the first consultation and the recommendation of a remedy met the operational goal of being within 48 h: 50.6% of cases were seen in the first day after contact, 32.6% were seen immediately, and a further 10.7% of cases were seen on the second day after contact with the clinic. Most cases (83.2%) were provided with remedy recommendations within 24 h.

Another measure of how team resources are most efficiently deployed is an analysis of the number of consultations conducted. In this instance, consultations included involved any exchange with a practitioner including emails, calls, and texts. Our findings show that the majority of cases were resolved in five or fewer consultations (61.36%), with a further 27.19% resolved after 6–10 clinic contacts.

No clear relationship was found between days ill before clinical intervention and the length of time between first intervention and the resolution of symptoms; in other words, not all clients that had the longest time to resolution (1.62% in range of 26–50 days) were ill longer before the clinical team’s intervention. Conversely, some clients who submitted a request for care early in their illness did not have their symptoms resolve quickly.

### 3.3. Measures Indicating Effectiveness of Treatment

Symptom improvement (Table 3) and/or resolution was cited by most clients, with 57.7% reporting full resolution, while some clients reported their symptoms as much better (14.4%) and somewhat better (11.5%). Conversely 5.1% of clients reported their symptoms as unresolved/unchanged, while no clients reported their symptoms unchanged, and very few reported that they were somewhat worse (1.1%) or much worse (0.3%). Some clients (7.5%) did not respond to questions about their status, and 0.7% of clients were referred out and did not receive services from HHN.

Another measure of clinical success is an analysis of responses to individual remedy prescriptions. Our findings, measured on a seven-point Likert scale, reveal the majority of individual remedy responses as somewhat better (44.4%) over the course of care. Other responses to an individual remedy taken showed 19.7% as much better, 12.4% as unchanged, and 9.6% as reporting symptoms as resolved. On the other hand, 8.8% of participants reported being somewhat worse, 0.5% as much worse, and 0.3% as unresolved. Additionally, it was found that 2.4% did not take the remedy recommended, and 1.9% showed no response from the client.

## 4. Discussion

This study highlights four discussion points, two of which are related to the practice of homeopathy itself and two of which address broader and more important implications for epidemic and pandemic capacity and readiness.

The findings of this study reveal that 81% of participants and users of this service did not have a working relationship with a practitioner of Homeopathy at the time of intake. This number fluctuated very little throughout the years of this study, ranging from 79.6% to 82.7%. At the very least, this would tend to confirm that very ill people were amenable to the utilization of a previously unknown modality and intervention in a time of crisis. In and of itself, this observation might make some sense if it was limited to the beginning of the pandemic, when conventional healthcare services were stretched and so few other healthcare options existed, but the trend continued well after COVID-19 was able to be managed by conventional medical services. Even in the face of widely disseminated public health services warnings about alternative/complementary COVID-19 treatments, users continued to take up the ‘unknown treatment approaches’ of homeopathy services at HHN. In the post-pandemic washup, many papers have contributed to a wide narrative about consumers using ‘non evidence based treatments and making erratic choices’ [18,19,20,21]; however, the experience of at those the HHN clinic found that, very often, clients were drawn to the service more as a response to a lack of clarity in the public health message [22,23,24] and poor health outcomes in conventional settings [25,26] as much as for any other reason. The very high percentage of users that did not report prior engagement with homeopathy also highlights that a large number of people were introduced to homeopathy during the pandemic. Subsequent research could be aimed at understanding more deeply the reasons for the uptake of homeopathy services at the time.

The second discussion point relates to the relevance of traditional methods of prescribing in homeopathy. The authors are mindful that the raw data from this clinical setting are not compared to any other similar COVID-19 data, and, as a consequence, we cannot over-reach and make any claims about clinical efficacy. That acknowledged, however, this study has shown that the data do clearly show some individual improvements following the use of specific remedies, but, more importantly, these data confirm the general effect of homeopathy as a treatment (inclusive of the effects of the consultation, the therapeutic relationship, and the effects of changing remedies over the course of treatment) [27]. These results were established using traditional Hahnemannian methods of prescribing, as opposed to new systems of prescribing or the employment of protocols of homeopathic remedies. During the pandemic, the clinical team gave many imperfect remedies out of necessity. Many clients ended up taking the second- or third-choice remedies because of accessibility and proximity to remedy sources, yet the HHN clinic continued to see excellent results. In the face of its detractors, traditional Hahnemannian homeopathy is shown to be effective in simply tracking the symptoms of the case, prescribing the best matched remedy perceived and working forwards to resolution. As such, future clinical research related to the clinical success of these other methods as compared with traditional Hahnemannian homeopathy seems warranted.

Thirdly, at the time when COVID-19 emerged, this clinical initiative, modest well managed, urgent care, telehealth system, was immediately inundated. At the time (March–December 2020), there were no clear treatments, and public health directives changed by the day. Perhaps most importantly, however, at the front line, desperate clients with dangerously low oxygen levels (in many cases in the low 80s) were being refused access to ambulances across the US as sick people were told they were not sick enough to receive any treatment in overcrowded hospitals. The clinical outcomes highlighted in this paper, when seen through the lens of the dire public health scenario, demonstrate that homeopathy can be a legitimate option in epidemic preparedness and symptom treatment. Homeopathy has a long lineage in the treatment of epidemics. From the early 1800s, homeopathy has been effectively utilized in the management of scarlet fever (Germany 1801, London 1854) [7,28], typhus (Leipzig 1813) [29], pneumonia (Paris 1847, Vienna 1850) [28], cholera (Paris 1849, London 1854) [28], diphtheria (Boston 1915) [28], Dengue [30] (Honduras 2007) [31], Leptospirosis (Cuba 2007) [32], and Chikungunya (Haiti 2015) [33], amongst many others. While it is clear that some case reports, case series, and outcome narratives of these front-line situations have not always been of the highest quality; en-masse, they represent a phenomenon which cannot be easily dismissed, especially in the light of the data presented herewith. As we have shown in this study, the outcomes of treating clients using the *genus epidemicus* theory of Hahnemann is demonstrably useful [6,34,35,36,37,38]. Homeopathy’s founder, Samuel Hahnemann, gave clear guidance on the importance of approaching each epidemic disease as different from all previous epidemics (*Aph 100*) [39]. Seen from within homeopathy, this framing reduces the likelihood of bias in the study of the emerging collective disease pattern. At the very least, a systematic review of epidemic studies in homeopathy is warranted. Further, Health Services Research and Public Health research projects are warranted to investigate the ways in which such necessary stop-gap clinical services could become implemented in early- and later-phase response to pandemics. Additional research using similar methods used in this study but compared with similar data from other conventional and complementary therapies is therefore suggested and deemed necessary.

There is an additional fourth, broader point to be made here. Our findings suggest that it was not only that a telehealth service employed a complementary therapy that was the source of the outcomes. There was also a collaborative team model employed. HHN’s collaborative, horizontal team structure was essential for the kind of fresh, open-minded approach needed to grapple with the serious and disconcerting symptoms of COVID-19 as well as finding ongoing solutions to the quality of care, privacy, and technical barriers and security challenges.

Moreover, HHN’s organizational model draws on a mutual aid structure, whereby dynamic, flexible systems are created that empower a community to meet emerging needs, especially when more formal (in this instance public health) structures are strained or failing. This horizontal structure fosters a collective critical thinking and decision-making process, the prevention of group conflict, and the management of burnout [40]. At the time, and especially in the early months of the pandemic marked by collective fear, confusion, and grief for so many suffering and dying, these specific structural attributes facilitated a level-headed study of the epidemiological disease symptom pattern. From the outset, this collaborative model was employed, and the team met and reviewed cases regularly and received supervision regularly, in doing so mitigating the effects of burnout [41,42,43,44,45,46].

There are limitations to be mentioned in this study. The objective of this study is to retrospectively analyze data collected during the pandemic. It is not—nor does it claim to be—a paper focusing on efficacy, meaningful statistical analysis, or a comparison of findings with conventional medical research to control confounding, limit bias, and increase the validity of findings. As such, advanced analysis was not conducted in this study. However, what we have aimed to achieve is to highlight the experience of one clinic providing homeopathy care to thousands of clients and describe and discuss outcomes. In this review of a pragmatic clinical situation, this research uses a convenience sample of clients who provided permission to have their data used.

## 5. Conclusions

The HHN team used traditional Hahnemannian homeopathic remedies and principles to make a difference in the lives of 3495 people during the COVID-19 pandemic. In doing so, they introduced a large majority of those people to Homeopathy for the first time. Further, they continued the long tradition of assisting in managing symptoms of clients in an epidemic using homeopathic medicine by employing a collaborative, integrative, and horizontal flexible structure. In a global climate punctuated more and more by socio-political upheaval, population displacement, climate change, technological inequalities, and vulnerabilities, alongside the successes and failures of conventional health care offerings, the authors present their findings of this study to stakeholders (public health officials, researchers, policy makers) to the consider the novel experiences and challenges faced by this HHN clinical team. Further research is urgently needed to explore the implementation and benefits of innovative, flexible healthcare structures, such as the one used in this study, that can meet the unpredictable and fluctuating public health needs in our changing world.

## Figures and Tables

**Table 1 healthcare-13-00003-t001:** Age, Gender, COVID-19 status at intake, and if there was an existing relationship with a homeopath.

Age: Minimum and Maximum		
Age	Youngest	Eldest
	<1	93
Age	Percent	Number
pediatric (0–17)	41.9%	1299
adult (above 18)	58.1%	1798
Gender	Percent	Number
Female	61.7%	1928
Male	32.4%	996
Fields left blank	5.7%	175
Declined to say	0.1%	9
Neither Male nor Female	-	<5
Transgender	-	<5
Grand Total	100.0%	3111
COVID-19 Status at Intake	Percent	Number
not tested, with COVID adjacent symptoms	36.7%	1283
not applicable to this case	32.5%	1136
positive at time of request	16.0%	561
COVID negative	6.7%	236
previously tested positive	3.5%	122
positive during time of care, after initial request	2.1%	72
Total Number		3410
Fields left blank		89
Grand Total		3499
Ever had contact with or have an existing Homeopath	Percent	Number
Yes	18.8%	585
No	81.2%	2525
Total Number		3110
Fields left blank	11.1%	389
Grand Total		3499

**Table 2 healthcare-13-00003-t002:** Days between request and intake and the number of client contacts to resolution.

**Days Between Request and Intake**	**Percent**	**Number**
<1	32.6%	359
1	50.6%	557
2	10.7%	118
3	2.5%	28
4	1.1%	12
5	0.5%	5
>5	2.27%	25
**Number of client contacts to resolution**	**Percent**	**Number**
1–5	61.36%	1817
6–10	27.19%	805
11–15	7.60%	225
16–20	2.13%	63
21–25	1.08%	32
26–30	0.34%	10
31–35	0.10%	3
36–40	0.10%	3
41–45	0.07%	2
46–50	0.03%	1

**Table 3 healthcare-13-00003-t003:** Outcome at Final Contact and Individual Remedy Response.

**Outcome at Final Contact**	**Percent**	**Number**
Resolved	57.7%	1735
Much Better	14.4%	433
Somewhat better	11.5%	347
Unresolved/Unchanged	5.1%	154
Somewhat worse	1.1%	33
Much worse	0.3%	10
Non-compliant	1.6%	48
No response from client	7.5%	225
Referred out	0.7%	21
**Individual Remedy Response**	**Percent**	**Number**
Resolved	9.6%	1317
Much Better	19.7%	2695
Somewhat Better	44.4%	6059
Unchanged	12.4%	1692
Somewhat Worse	8.8%	1202
Much Worse	0.5%	71
Unresolved	0.3%	41
Did not take the remedy	2.4%	327
No Response from client	1.9%	253

## Data Availability

The data that supports the findings of the study are available from the corresponding author (A.C.G.) upon reasonable request.

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
