# Peer review of "Challenges and Lessons from an Acute Telehealth Homeopathy Service During the Pandemic: A Case Series Exploring How Changing Demographics, Efficiency and Outcomes, Point to New Options for Epidemic Readiness"

_healthcare, 2024, doi:10.3390/healthcare13010003_

Round 1

Reviewer 1 Report

Comments and Suggestions for Authors

1. Avoid using the very short centences

2. "Abstract: In 2020, HOHM Foundation launched Homeopath....." the first time write the HOHM in full

3. Enhance 2.1 and 2.2 and add citations

4. Page 4 check the age category "pediatric 0-17" do we have human with 0 age?

5. Is there any reason to include those two categories? Neither Male nor Female and Transgender?

6. Enhance the discussion with literature review 

7. Update the references 

Comments on the Quality of English Language

Acceptable

Author Response

Reviewer 1

  1. Avoid using the very short sentences.

We thank the reviewer for this well-made observation. We have edited the entire manuscript removing any sentences considered short.

  1. "Abstract: In 2020, HOHM Foundation launched Homeopath..." the first time write the HOHM in full

Thanks for this comment. HOHM is not an acronym. It is written in full as it is.

  1. Enhance 2.1 and 2.2 and add citations

We thank the reviewer for this comment and have added citations to the manuscript in line with this suggestion.

  1. Page 4 check the age category "pediatric 0-17" do we have human with 0 age?

We thank the reviewer for this comment. On examination of the broader literature and looking at the conventions used in health services research and public health research, for example, Jones, S. H., St. Peter, C. C., & Ruckle, M. M. (2020). Reporting of demographic variables in the Journal of Applied Behavior Analysis. Journal of applied behavior analysis, 53(3), 1304-1315, infants between the age of zero and one, in other words in their first year of life, are identified statistically as beginning at zero.

  1. Is there any reason to include those two categories? Neither Male nor Female and Transgender?

We thank the reviewer for this comment. The authors have decided to maintain the use of this language, as there were two participants that identified as neither male nor female, and one participant that identified as transgender. In this regard, the authors are simply repeating the language related to gender used by participants in the study. We think it is important that all individual are represented. We do not want to exclude anyone, even while protecting the anonymity of the data by excluding the exact figures.  

  1. Enhance the discussion with literature review 

We thank the reviewer for this comment. It is assumed that the reviewer is speaking about the literature relating to clinical outcomes in a Homeopathy telehealth outpatient service. We have trawled the academic as well as the grey literature and made comments and added referencing changes as a consequence. If the reviewer was speaking of a literature review relating to the third discussion point, the use of homeopathy and epidemic disease diseases we have made some alterations to the manuscript that can be found lines 47-55.

  1. Update the references 

We thank the reviewer for this comment and have updated the references as requested and as appropriate.

Reviewer 2 Report

Comments and Suggestions for Authors

It is a great work. However, I have a comment are as follows to improve the work:

I was wondering if respectful authors consider related MeSH (Medical Subject Headings) for their study.

For example, I recommend to use “complementary and alternative medicine” instead of complementary medicine: 1- in the Abstract text, 2- Keywords, and 3- manuscript text.

Author Response

Reviewer 2

  1. I was wondering if respectful authors consider related MeSH (Medical Subject Headings) for their study.

We thank the reviewer for this comment. Our understanding is that the MeSH guidelines are designed for use by the National Library of Medicine for indexing and searching inside databases for journal citations as well as other data. This enables retrieval systems, such as NLM's PubMed, to provide subject searching of the data.  It is used in scoping and systematic reviews, and we have used it in those kinds of articles previously. But in this instance the objective of the paper was to report of clinical outcomes and discuss the implications on Human Health Before, During, and After COVID-19 and so we chose rather than a literature review to focus providing a collection of high-quality data and new insights into COVID-19 – as per the brief.

  1. For example, I recommend to use “complementary and alternative medicine” instead of complementary medicine: 1- in the Abstract text, 2- Keywords, and 3- manuscript text.

We thank the reviewer for this comment. The authors have consulted and our role in agreement that the use of ‘alternative’ in the acronym complementary and alternative medicine (CAM) has almost universally been dropped from the academic scholarly landscape of the recent research that we are familiar with. Even in this article we see complementary medicines being used in an integrative way. Hence the use of the term complementary medicine in this paper. That said, if the reviewer is adamant, then the authors will acquiesce and use CAM rather than CM.

Reviewer 3 Report

Comments and Suggestions for Authors

Review Statement:

Thank you for the review opportunity.
The research paper, titled "Challenges and lessons from an acute telehealth homeopathy service during the pandemic: how changing demographics, efficiency, and outcomes point to new options for epidemic readiness," provides a thorough examination of the strengths and weaknesses of telehealth homeopathy services during the COVID-19 pandemic. 
This study stresses the adaptability and endurance of telehealth services in meeting patient requests during a crisis, using an example from the United States. It also identifies crucial challenges, such as demographic trends and inefficiencies, which offer valuable insights into future pandemic preparedness.

Following a thorough assessment, I would recommend this paper for "Accept after minor revisions."

Comments for revision: The paper is well-written and offers a thorough study of the issue. However, I propose the following minor revisions:
1)The authors should define the methods used for data collection and analysis. (Chapter 2 should be expanded with detailed sources (setting, sampling, data input, data analysis, and so on), or the materials section should be combined into a single integrated component.
2)The authors should provide further particular examples of the demographic shifts you've identified via the sample cases.
3)The authors should explain various solutions to the efficiency difficulties stated. (Technical barriers, quality of care, privacy, and security perspectives)

Overall, this research makes substantial contributions to the area and offers practical suggestions for improving telehealth services during future pandemics.

Author Response

Reviewer 3

Thank you for the review opportunity. The research paper, titled "Challenges and lessons from an acute telehealth homeopathy service during the pandemic: how changing demographics, efficiency, and outcomes point to new options for epidemic readiness," provides a thorough examination of the strengths and weaknesses of telehealth homeopathy services during the COVID-19 pandemic. 

This study stresses the adaptability and endurance of telehealth services in meeting patient requests during a crisis, using an example from the United States. It also identifies crucial challenges, such as demographic trends and inefficiencies, which offer valuable insights into future pandemic preparedness.

Following a thorough assessment, I would recommend this paper for "Accept after minor revisions."

Comments for revision: The paper is well-written and offers a thorough study of the issue. However, I propose the following minor revisions:
1) The authors should define the methods used for data collection and analysis. (Chapter 2 should be expanded with detailed sources (setting, sampling, data input, data analysis, and so on), or the materials section should be combined into a single integrated component.

We thank the reviewer for this comment. We have amended the manuscript in line with this suggestion and have added where relevant details relating to setting sampling data input analysis. We have not combined all of the sections of the materials section into a single integrated component because of the comments that are contrary from another reviewer. However we have revised and re-written this section accordingly. Please see lines 74-118.

2) The authors should provide further particular examples of the demographic shifts you've identified via the sample cases.

We thank the reviewer for these comments. This paper identifies demographic features of participants, but it’s not really focused on shifting demographics.

3) The authors should explain various solutions to the efficiency difficulties stated. (Technical barriers, quality of care, privacy, and security perspectives)

We thank the reviewer for the specific comments and have amended the manuscript accordingly. Explanations as to the difficulties experienced in relation to efficiency are now to be found in both the Results section (3.2 Efficiency) and the Discussion section (fourth point). Please see page 6 beginning at line 149 and page 10 and line 254.
Overall, this research makes substantial contributions to the area and offers practical suggestions for improving telehealth services during future pandemics.

Reviewer 4 Report

Comments and Suggestions for Authors

Thanks for submitting the manuscript to the journal. I think several points should be cleared in the methodology of the manuscript.  

1. What were the complaints of the patients? and what were the treatments you offered?

2. When and how did you ask about the results of treatment?

3.Was there any sampling strategy?

Author Response

Reviewer 4

Thanks for submitting the manuscript to the journal. I think several points should be cleared in the methodology of the manuscript.  

  1. What were the complaints of the patients? and what were the treatments you offered?

We thank the reviewer for this comment. On reflection, the authors feel comfortable that there is clarity that the paper is about the treatment of COVID-19 and that the intervention is Homeopathy. If the author is talking about the specific remedies that were employed then the authors would argue, that is a different paper and outside of the purview of the edition of the specific journal relating to human health before during and after COVID-19. If the reviewer is interested in specific Homeopathic interventions, then the research team have published papers outlining several of these points as follows.

  1. When and how did you ask about the results of treatment?

We thank the reviewer for this comment and have made clearer to the reader how data was collected about the results of the intervention. Please see the amended manuscript at page 2.3. Data Collection line 86.

  1. Was there any sampling strategy?

Many thanks to the reviewer for this comment. We examined the entirety of our data set for purposes of the research presented here.  Additional work is being done on samples of the data based on symptoms and outcomes to be published in a separate work.

Reviewer 5 Report

Comments and Suggestions for Authors

This paper provides perspective on complementary practices and the real world impact of homeopathic medicines. 

The introduction is clear. It provides background on the historical use of homeopathic medicines and the more recent impact to reduce disease burden during the pandemic. 

The methods are presented clearly. However, the data collection for the outcomes could be considered subjective. 

For figures 1, 3, 4, 5, and 6, the axis could be changed to 100% to show a consistent schematic for comparison between the figures. 

The results support the methodology. 

The discussion is detailed. The authors describe the relevance of traditional medicines and the impact of COVID on public health initiatives. Although the data presented cannot be compared to other clinical study data, the potential future implementation of traditional medicines to treat diseases has been discussed. 

Author Response

Reviewer 5

This paper provides perspective on complementary practices and the real world impact of homeopathic medicines. 

  • The introduction is clear. It provides background on the historical use of homeopathic medicines and the more recent impact to reduce disease burden during the pandemic. 

We thank the reviewer for this comment.

  • The methods are presented clearly. However, the data collection for the outcomes could be considered subjective. 

We thank the reviewer for this comment.

  • For figures 1, 3, 4, 5, and 6, the axis could be changed to 100% to show a consistent schematic for comparison between the figures. 

We thank the reviewer for this comment. The figures have all been amended.

  • The results support the methodology. 

We thank the reviewer for this comment.

  • The discussion is detailed. The authors describe the relevance of traditional medicines and the impact of COVID on public health initiatives. Although the data presented cannot be compared to other clinical study data, the potential future implementation of traditional medicines to treat diseases has been discussed

We thank the reviewer for this comment.

Reviewer 6 Report

Comments and Suggestions for Authors

I have gone through and reviewed the article titled “Challenges and lessons from an acute telehealth Homeopathy service during the pandemic: How changing demographics, efficiency and outcomes, point to new options for epidemic readiness” with the great interest. I appreciate the authors for the good attempt. However, there are numerous concerns present in the manuscript.

  1. Title looks attractive, but author may consider following appropriate checklist (STROBE, etc) while making title.
  2. Abstract: Word count exceeding the healthcare, MDPI requirements. Also, the critical aspect of study design is missing. Moreover, it would be better to give analytical results than a simple description.
  3. Introduction: Interesting to see the introduction of the manuscript has a high level of plagiarism of their previous work (https://www.liebertpub.com/doi/10.1089/imr.2024.0014). If the introduction is the same, there is no need for this study. The discussion around alternative treatments during overwhelmed healthcare conditions is crucial, but it could benefit from a more critical view of its comparative role to conventional medicine.
  4. Introduction: Justification is not sufficient (considering authors’ recent publication contain the same introduction).
  5. Introduction: A clear and consiced statement/objectives must be provided.
  6. Methods: Please follow the checklist or guidelines as per the study. Authors need to be more specific on it.
  7. Methods: Similar to the introduction, there are hugely similar contents from the authors previous paper. Is the first study salami slicing ? I see the first study as an audit.
  8. Authors need to explain more about data analysis. Actually, the data analysis section is not data analysis according to the content. Furthermore, the authors used simple descriptive data analysis. Considering the rich nature of data set, authors need to go for multivariate analysis. Otherwise, the findings may not be sufficient for publication in prestigious journal.
  9. Results: Kindly do not make overstatements about results such as improvement in symptoms (as the authors do not have a control).
  10. Discussion – I consider the authors made attempt. However, the discussion would benefit from a comparison with outcomes in conventional telehealth settings during the pandemic, which would provide a balanced view. Furthermore, the authors must revise the discussion according to higher levels of analysis and results.
  11. Please emphasize the limitations of the study.
  12. The conclusion could more strongly relate findings to practical implementation in epidemic readiness frameworks such as COVID-19.

Author Response

Reviewer 6

1. Title looks attractive, but author may consider following appropriate checklist (STROBE, etc) while making title.

Many thanks to the reviewer for this comment. The authors have reflected on the STROBE guidelines. We feel that the title reflects the data collection and analysis, as well as the discussion and conclusions in this regard.

2. Abstract: Word count exceeding the healthcare, MDPI requirements. Also, the critical aspect of study design is missing. Moreover, it would be better to give analytical results than a simple description.

We have described the study design in the paper in the abstract. The word count is 283 words (similar to the range of abstracts found published in this journal (between 253-296 words). As no analytical results were discussed in the paper we have stuck to descriptive results in the abstract.

3. Introduction: Interesting to see the introduction of the manuscript has a high level of plagiarism of their previous work (https://www.liebertpub.com/doi/10.1089/imr.2024.0014). If the introduction is the same, there is no need for this study. The discussion around alternative treatments during overwhelmed healthcare conditions is crucial, but it could benefit from a more critical view of its comparative role to conventional medicine.

Many thanks to the reviewer for this comment. The extent of overlap in the introduction with a previous publication about an audit of the clinic is well made, and, as a consequence of this oversight, the abstract, introduction and sections of the results section have been entirely rewritten.

4. Introduction: Justification is not sufficient (considering authors’ recent publication contain the same introduction).

We thank the reviewers for this observation and, as above, have re-written the introduction, including the justification for the study.

5. Introduction: A clear and consiced statement/objectives must be provided.

Many thanks for this comment from the reviewer. The manuscript has been amended accordingly with a clear and concise statement of objectives. See line 65-71.

6. Methods: Please follow the checklist or guidelines as per the study. Authors need to be more specific on it.

We thank the reviewer for this comment and have been through all of the checklists and guidelines for the journal as well as the STROBE guidelines above and amended the manuscript accordingly.

7. Methods: Similar to the introduction, there are hugely similar contents from the authors previous paper. Is the first study salami slicing ? I see the first study as an audit.

Many thanks to the reviewer for this comment. As identified above, with the rewriting of the introduction, results, and methods sections, this research paper is its own entity. As a result the reviewers comment about salami slicing of the same data is not relevant here. It is a different data set, and the objective of the papers are entirely different. This paper describes a much larger data set, n=3499 in contrast to the audit paper in which with the n= 300 participants analyzed against standards.

8. Authors need to explain more about data analysis. Actually, the data analysis section is not data analysis according to the content. Furthermore, the authors used simple descriptive data analysis. Considering the rich nature of data set, authors need to go for multivariate analysis. Otherwise, the findings may not be sufficient for publication in prestigious journal.

Many thanks to the reviewer for this comment. The intent of this research was to provide a big picture, descriptive view of the work that had taken place during and after the heights of the COVID-19 pandemic. There is a breadth of data here that is being examined using advanced statistical analysis and will be available in the future. Our focus on the broad-stroke approach to the analysis is to provide a foundation of what happened, to tell the story, and emphasize the people aspects of the pandemic.

9. Results: Kindly do not make overstatements about results such as improvement in symptoms (as the authors do not have a control).

Many thanks to the reviewer for this comment. Please see the amended manuscript lines 214-220.

10. Discussion – I consider the authors made attempt. However, the discussion would benefit from a comparison with outcomes in conventional telehealth settings during the pandemic, which would provide a balanced view. Furthermore, the authors must revise the discussion according to higher levels of analysis and results.

Many thanks to the reviewer for this comment. The authors are mindful that the entire point of this edition of the journal was not about outcome comparisons.

It has been more than three years since COVID-19 was declared a global pandemic. Since that point, we have experienced profound changes in the ways that we work, socialise, and learn. The severity of this transformation has allowed us to examine daily practices, social norms, institutions, and the positive and negative aspects of our former lifestyles. The pandemic compelled us to appreciate the precious details of our lives that we have taken for granted: workplaces, human touch, parties, travel, and access to and the utilisation of healthcare facilities. We are thus able to reflect on the ways we live now in order to modify our cultures and find different ways to improve the quality of life for future generations. Many trends that had already started before COVID-19 have been accelerated by the impact of the pandemic. Examples include the digital economy, with the rise of telemedicine in the healthcare delivery services. One of the hardest things to deal with in this period was maintaining physical distance. Physical interactions are an essential part of human social experience, and they are particularly important for the development of young people. Social distancing, school closures, and physical isolation from friends have been especially challenging for adolescents. Some have subsequently experienced feelings of loneliness, hopelessness, and sadness. Remote learning and homeworking under lockdown have also impacted the wellbeing of university students and young workers. Over the course of the pandemic, people of all ages reported symptoms consistent with anxiety and depression, regardless of severity of any viral infection. Additionally, increasing evidence of post-COVID long-term effects has been reported in the literature. Long COVID describes a range of symptoms, including fatigue, dyspnea, brain fog, and mental health disorders. COVID is becoming endemic, which means that we will live with it as we do with the flu, i.e., without consequences as severe as those seen in the first three years. However, while some countries have reported a decline in COVID-19 cases and deaths, largely due to high vaccination coverage, there is still a relevant public health concern about refugees and unvaccinated people in many developing countries. The behaviour we all exhibit and the epidemiology of the virus itself are extremely important. New SARS-CoV-2 variants show increasing levels of concern and are likely to impact the epidemiological situation worldwide.

With respect to the point made the reviewer, this work is to retrospectively analyze the data collected in the pandemic. It is not – nor claims to be a paper focused on efficacy or meaningful statistical analysis or comparison of findings from conventional medical research. It is the experience of one clinic using thousands of clients and their outcomes. In this review of a pragmatic clinical situation, there was not an opportunity to have a prospective study design at the outset. It is a convenience sample or clients who provided permission to have their data to be used.  In summary, in response to this excellent point made by the reviewer, the revisions suggested would require significant more research and a re-examination of the work from a vastly different perspective. At this juncture, work comparing clinical result is underway by our research team to analyze the data form these different perspectives but is not the focus or intent of this paper.

10. Please emphasize the limitations of the study.

Many thanks to the reviewer for this comment. The limitations section has been added accordingly. See line 266.

11. The conclusion could more strongly relate findings to practical implementation in epidemic readiness frameworks such as COVID-19.

Many thanks to the reviewer for this comment. Please see the response to reviewer 6 in point 10 above.

Round 2

Reviewer 4 Report

Comments and Suggestions for Authors

.

Author Response

We note the recommendation of the reviewer.

Reviewer 6 Report

Comments and Suggestions for Authors

 I appreciate the authors' tremendous and commendable efforts in revising the manuscript.

However, still, I have some concerns:

1. I understand the explanation given by the authors regarding the title. However, I suggest looking at the STROBE Guidelines https://www.strobe-statement.org/checklists/ 

1. (a) Indicate the study’s design with a commonly used term in the title or the abstract

2. Again, I found methods can be enhanced in the order of presentation with the STROBE guidelines.

3. As authors must be aware, advanced analysis is done to control confounding, limit bias, and increase the validity of findings. If the authors could not complete it for some reason, it must be explicitly explained in the limitations. Therefore, the readers will not misinterpret the findings.

I wish you all the best, and congrats once again for making great efforts to revise the manuscript to enhance its quality.

Author Response

Reviewer 6

I appreciate the authors' tremendous and commendable efforts in revising the manuscript.

However, still, I have some concerns:

  1. I understand the explanation given by the authors regarding the title. However, I suggest looking at the STROBE Guidelines https://www.strobe-statement.org/checklists/ 

(a) Indicate the study’s design with a commonly used term in the title or the abstract

Response: Many thanks for your feedback in this regard. The authors have carefully studied the STROBE guidelines and have found it difficult to determine exactly which guideline you wish us to use. As a decision needed to be made, we’ve used the ‘case control study guidelines’, even though this study is not nor has ever been a case controlled study. However, it is the closest guideline given the possibilities. Therefore we have changed this title from this…

Challenges and Lessons from An Acute Telehealth Homeopathy Service During The Pandemic: How Changing Demographics, Efficiency and Outcomes, Point to New Options for Epidemic Readiness

To this

Challenges and Lessons from An Acute Telehealth Homeopathy Service During The Pandemic: A Case Series Exploring How Changing Demographics, Efficiency and Outcomes, Point to New Options for Epidemic Readiness

In the Abstract we have amended this sentence from this…

Methods: in this pragmatic study, cases of positively testing or probable COVID-19 (n=3,495) are analyzed using conventional quantitative analysis.

To this…

Methods: in this pragmatic case series study, cases of positively testing or probable COVID-19 (n=3,495) are analyzed using conventional quantitative analysis.

  1. Again, I found methods can be enhanced in the order of presentation with the STROBE guidelines.

Response: We thank the reviewer for this comment. As a consequence, we have amended the text so that the sections of the method section are an alignment with the strobe guidelines. See line 73-127.

  1. As authors must be aware, advanced analysis is done to control confounding, limit bias, and increase the validity of findings. If the authors could not complete it for some reason, it must be explicitly explained in the limitations. Therefore, the readers will not misinterpret the findings.

Response: We thank the reviewer for this comment. As a consequence, the text has been amended in the limitations section. Please see line item 267-273.
